# A Missense Variation in *PHACTR2* Associates with Impaired Actin Dynamics, Dilated Cardiomyopathy, and Left Ventricular Non-Compaction in Humans

**DOI:** 10.3390/ijms24021388

**Published:** 2023-01-10

**Authors:** Pierre Majdalani, Aviva Levitas, Hanna Krymko, Leonel Slanovic, Alex Braiman, Uzi Hadad, Salam Dabsan, Amir Horev, Raz Zarivach, Ruti Parvari

**Affiliations:** 1The Shraga Segal Department of Microbiology, Immunology & Genetics, Faculty of Health Sciences, Ben-Gurion University of the Negev, Beer-Sheva 84105, Israel; 2The National Institute for Biotechnology in the Negev, Marcus Campus, Ben-Gurion University of the Negev, Beer-Sheva 84105, Israel; 3Department of Pediatric Cardiology, Soroka University Medical Center, Faculty of Health Sciences, Ben-Gurion University of the Negev, Beer-Sheva 84101, Israel; 4The Ilse Katz Institute for Nanoscale Science and Technology, Marcus Campus, Ben-Gurion University of the Negev, Beer-Sheva 84105, Israel; 5Pediatric Dermatology Service, Soroka University Medical Center, Beer-Sheva 84101, Israel; 6Department of Life Sciences, Faculty of Natural Sciences, Ben-Gurion University of the Negev, Beer-Sheva 84105, Israel

**Keywords:** dilated cardiomyopathy, left ventricular non-compaction, *PHACTR2*, actin monomer pool, actin polymerization, cell movement

## Abstract

Dilated cardiomyopathy (DCM) with left ventricular non-compaction (LVNC) is a primary myocardial disease leading to contractile dysfunction, progressive heart failure, and excessive risk of sudden cardiac death. Using whole-exome sequencing to investigate a possible genetic cause of DCM with LVNC in a consanguineous child, a homozygous nucleotide change c.1532G>A causing p.Arg511His in *PHACTR2* was found. The missense change can affect the binding of PHACTR2 to actin by eliminating the hydrogen bonds between them. The amino acid change does not change PHACTR2 localization to the cytoplasm. The patient’s fibroblasts showed a decreased globular to fibrillary actin ratio compared to the control fibroblasts. The re-polymerization of fibrillary actin after treatment with cytochalasin D, which disrupts the actin filaments, was slower in the patient’s fibroblasts. Finally, the patient’s fibroblasts bridged a scar gap slower than the control fibroblasts because of slower and indirect movement. This is the first report of a human variation in this *PHACTR* family member. The knock-out mouse model presented no significant phenotype. Our data underscore the importance of *PHACTR2* in regulating the monomeric actin pool, the kinetics of actin polymerization, and cell movement, emphasizing the importance of actin regulation for the normal function of the human heart.

## 1. Introduction

Dilated cardiomyopathy (DCM) is a leading cause of heart failure and the predominant type of cardiomyopathy, affecting approximately 1 in 400 people [1]. Many known causes include infectious agents, drugs or toxins, peripartum stress, nutritional deficiencies, and autoimmune disorders [2]. However, approximately one-half of all cases are idiopathic; among them, the estimated familial DCM is 30–50% [3]. It is inherited mainly in an autosomal dominant (AD) pattern [4] and usually presents in the second or third decade of life [3,5]. Mutations in titin, lamin A/C, and myosin heavy chain (MHC7) account for over 30% of genetic DCM [2]. Other mutated genes affect cardiomyocyte’s ultrastructure, sarcomeric integrity, force generation, cellular metabolism, and transcriptional regulation [6]. Autosomal-recessive (AR) mutations leading to DCM are far less common. They were reported in genes encoding cardiac structural proteins, proteins transmitting the generated force, cardiolipin metabolism, chaperones, RNA binding, and ATP production [7]. In a study of 350 DCM patients, it was found that the patient age was significantly younger in the AR form of DCM compared to the AD form, with a more rapid progression to death or heart transplant, all of which were under 21 years of age [5]. Left ventricular non-compaction (LVNC) often represents a phenotypic variation in DCM. A sizeable genetic overlap of LVNC with DCM supports the hypothesis that many LVNC cases are a variable morphological phenotype of an underlying cardiac disease and identify a distinct genetic etiology in a subset of cases [8]. The factors that cause patients with DCM-causing variants to develop or present with LVNC are unknown [8]. A study on the genetic contribution to primary pediatric cardiomyopathy, performed by the exome sequencing of 528 children with cardiomyopathy, demonstrated that less than 20% of the causative mutations were identified in pediatric DCM patients and illustrated differences in the genetic architecture between age, ancestry, and sub-phenotype [9]. Reclassification was suggested as a need in pediatric DCM [6].

DCM is associated with a loss of myofibrils and sarcomeric disorganization [10]. The precise control of the actin filament length contributes to the proper function of the contractile apparatus. Filament growth is affected by the G-actin/F-actin equilibrium, which is regulated by G-actin sequestering CAP proteins [11]. *PHACTR2* is a member of a family of four phosphatases and actin regulator (PHACTRs) proteins that exhibit the strong modulatory activity of protein phosphatase 1 (PP1) and an actin-binding activity [12]. *PHACTR2* is ubiquitously expressed; its expression in the heart is twice its expression in skeletal muscle (GTEx portal), and it was found to interact with cardiac actin [13]. *PHACTR1* is the only *PHACTR* family member associated with a human phenotype. Three de novo mutations in the *PHACTR1* gene were found to be associated with developmental and epileptic encephalopathy (OMIM *608723). PHACTR proteins were suggested to be involved in human diseases. A study on the genome-wide association of early-onset myocardial infarction reported that one SNP out of eight loci reached genome-wide significance in an intron of *PHACTR1* [14]. The same SNP was also associated with coronary heart disease (CHD)-susceptibility and coronary artery stenosis in type 2 diabetes [15]. Another study reported an SNP in the *PHACTR2* locus with genome-wide significance for Parkinson’s disease [16]. A missense variation causing p.D454H in *PHACTR2*, together with missense variations in *TTN* and *NOS3*, was suggested to be associated with primary platelet secretion defects of a bleeding tendency [17]. Finally, *PHACTR4* was proposed to act as a tumor suppressor [18]. A forward genetic screen for embryonic defects in mice identified a missense mutation in the PP1-binding motif of Phactr4. This mutation specifically disrupts the binding of PP1 to Phactr4 and causes the misregulation of PP1 activity [19]. Phactr4 mutant embryos display defects in neural tube closure and eye development due to abnormal cell proliferation controlled through the Phactr4-PP1-Rb (Rb: retinoblastoma protein) pathway [19].

The present study demonstrates a homozygous missense variation in *PHACTR2* that associates with DCM and LVNC in a highly inbred family of Bedouin ancestry. The study of the patient’s fibroblasts suggests that the patient’s cardiac phenotype may be caused by the perturbation of the globular actin pool and the disturbed kinetics of actin dynamics by the homozygous missense variation in *PHACTR2*.

## 2. Results

### 2.1. Clinical Findings

We recruited a Bedouin family with one patient; the parents were first-degree cousins. The mother’s fetus was diagnosed with a dilated left ventricle and decreased left ventricular function with normal cardiac anatomy at 32 weeks of gestation. He had a normal childbirth without the need for any medical intervention or other abnormal events to mention. At ten weeks of age, the patient presented with severe signs of heart failure, respiratory distress, a failure to gain weight, and feeding intolerance. His chest X-ray demonstrated cardiomegaly with pulmonary edema, and the first echocardiography study showed a severely dilated left ventricle with reduced function. The left ventricle structure presented non-compaction with normal coronary arteries and no valvular abnormalities. His blood pressure was within the normal range. The left ventricular and intraventricular septum were thin. The lateral wall and apex of the left ventricular were characterized by a two-layered myocardium composed of a non-compacted and compacted layer. The compacted layer’s thickness was 3.2 mm, and the non-compacted thickness was 13 mm. Nasal swabs and serology for viruses that may cause myocarditis were negative. Serum troponin testing was negative. A diagnosis of dilated cardiomyopathy (DCM) was established (Table 1) based on the British Society of Echocardiography criteria [20]. The patient improved after intense treatment with IV inotropes and IV Furosemide. However, after his initial improvement, he required oral furosemide, an ACE inhibitor, and carvedilol. During follow-up, his echocardiographic studies have been mostly stable. Presently, in his teens, he continues with his oral medication treatment (furosemide, ACE inhibitor, and carvedilol), yet he complains of chronic fatigue and exercise intolerance. His intelligence growth and development parameters are normal; he has no complaints about wound healing or other medical issues.

All individuals in the patient’s nuclear family underwent a complete evaluation of their hearts. No abnormalities were identified in any individual except the DCM patient (Appendix A). They all had normal developmental milestones.

### 2.2. Identification of the PHACTR2 Variation

Although the patient’s parents and five siblings were healthy, we verified the possibility of a causative heterozygous variation. A heterozygous variation could cause DCM in the case of compound heterozygous, a de novo, or a low-penetrance variation. First, we looked into the genes in which mutations were reported to cause DCM. We looked for variations in the exome results of patient II-3 in 47 genes reported to have mutations causing familial and non-familial DCM (Appendix A) [9,21,22]. We identified three missense heterozygous variations with less than 0.1% allele frequencies in the public databases (gnomAD browser, 1000 Genomes, ExAC, and EVS) in these genes in patient II-3. All three variants were negated by the segregation analysis of at least four healthy family members presenting variation in heterozygosity, which was performed by the PCR amplification of the variation’s DNA, followed by Sanger sequencing (Appendix A). Second, we proceeded to verify all other possible causative heterozygous variations. We found 8624 heterozygous variations in the patient exome. Only 135 had an allele frequency of less than 0.1% in the public databases mentioned above and less than 0.5% in our internal Bedouin laboratory Exome database. None of these variations were identified in the same gene, excluding the possibility of compound heterozygous as causative. Assuming a de novo variation, we negated all variations reported in the public databases- gnomAD and GeniePool, which mostly represented the Bedouins population, leaving 37 remaining variations. We continued the negation process based on the benign aggregated prediction score by the Franklin browser, variations in genes associated with different clinical presentations, and finally, excluded three remaining variations in genes with a low expression in the cardiac tissue (Appendix A). Thus, probably none of these variations represent a de novo, possibly causative variation.

Assuming homozygosity by the descent of a recessive mutation as the likely cause of the disorder, we analyzed the exome sequence of patient II-3. We first excluded the possibility of missing a whole exon by analyzing the WES coverage. Next, we identified 81 homozygous variations with less than 0.1% allele frequencies in the public databases (gnomAD browser, 1000 Genomes, ExAC, and EVS). A total of 69 out of the 81 variations were ruled out by our internal laboratory exome database of the Bedouin population. Five out of the 12 remaining variations were negated due to the non-cardiac expression of the genes and low damage prediction scores (CADD < 10, Omicia Variant Score < 0.2, SIFT Score > 0.1), as detailed in Appendix A. The segregation in the family for the seven remaining variants was verified by the PCR amplification of the DNA containing the variation, followed by Sanger sequencing. Two were segregated as expected (Appendix A). *WDR60* encodes a ciliary intraflagellar transport component. Variations in *WDR60* were associated with Jeune asphyxiating thoracic dystrophy with or without polydactyly [23,24] or retinal degeneration and polydactyly [25], excluding the variant in *WDR60* as the causative variant. The only remaining likely causative variation was on chromosome 6:144095295 G to A (GRCh37/hg19) in exon 8 of the phosphatase and actin regulator 2 genes (*PHACTR2*, Gene ID: 20956; NM_014721.3, c.1532G>A), causing amino acid replacement p.Arg511His. This missense variant had high damaging prediction scores by all predictions (SIFT: 0.025, CADD: 32, PolyPhen: damaging, and Omicia: 0.925), indicating the probability that the variant was deleterious. According to the American College of Medical Genetics (ACMG) and Genomics guidelines [26], *PHACTR2* gene variation c.1532G>A is classified as a variant of uncertain significance (VUS). The variation had an extremely low frequency in the gnomAD population databases (moderate pathogenic evidence, PM2). Regarding in silico prediction for the missense variant, computational prediction tools unanimously supported a benign effect on the gene (benign supporting evidence, BP4). There was no clinical evidence for this variant using ClinVar classification. Its segregation in the family shows that the DCM patient is the sole homozygous member (Figure 1A,B).

The variant was not present in our collection of Bedouin exomes of 780 individuals who were not cardiac patients. It was also not reported in the database of 77 Bedouins exomes [27], in the Qatari genome of more than 1000 exomes with a majority of the Bedouin population [28], and in the Franklin community of more than 150,000 exomes consisting of both healthy and affected individuals [29]; thus, its prevalence in the Bedouin population is less than 1 in 1714 chromosomes. Its total allele frequency in the gnomAD database is 0.00003611, with no homozygotes in any population.

### 2.3. Effect of the Candidate Variation on the Structure of PHACTR2 Protein

The RPEL repeat domains mediate the binding of unpolymerized actin, and the interaction protein phosphatase 1 (PP1) is mediated by the highly conserved C-terminal tail [12,19,30,31]. The candidate variation is located in the second of four RPEL repeats that are responsible for the interaction between PHACTR2 and actin [32] (diagram, Figure 1C) (numbered as RPLE1, the first of the three carboxy-terminal domains for which the structure was solved [32]). Arg 511 is conserved both evolutionarily, down to zebrafish, and functionally in all PHACTR family members (Figure 1C). We verified the effect of the candidate variation on the experimental crystal structure of mouse PHACTR1, which is conserved between PHACTR1 and PHACTR2 at the variant location site since there is no experimental structure of PHACTR2. The structure of mouse PHACTR1 with actin revealed that three G-actins surrounded the crank-shaped RPEL2 domain, forming a closed helical assembly. The PHACTR1 C-terminal triple RPEL repeat forms only a trivalent actin RPEL complex [32]. Since the residues around the mutation site are conserved, we used the crystal structure of mouse PHACTR1 with actin for the structural analysis (PDB:4B1Z, numbers are presented for humans for simplicity). In the PHACTR1 structure, Arg 511 creates hydrogen bonds and charged interactions with Glu 504 and Asp 505 (both glutamates in mice). These interactions stabilize the fold at the end of RPEL2 [32] (Figure 1D). The replacement of Arg at position 511 by His eliminates both the charge and hydrogen bond interactions and most likely destabilizes the end of RPEL2. Since RPEL2 interacts with actin molecules from both sides [32], the misfolding of the RPEL2 end is predicted to disrupt the whole complex.

### 2.4. Subcellular Distribution of PHACRT2 Is Not Affected by the Change in Arg 511 to His

PHACTR members localize to different subcellular compartments: the lamellipodium, focal adhesions, the cytoplasm, and nucleolus [33]. Rho-actin signaling induced by serum stimulation promotes the nuclear accumulation of Phactr1 but not other Phactr family members [34]. In unstimulated conditions, all four Phactr proteins showed a very similar distribution, with most cells scoring as diffusely localized. Phactr2 appeared mainly in the cytoplasm, and Phactr1 and Phactr3 displayed slightly higher nuclear distribution than the other family members [31]. In resting cells, Phactr4 and Phactr3 (also known as Scapinin) localized to the plasma membrane and were targeted there by the N-terminus, which is highly conserved in all members of the Phactr family [31,35]. The induction of phactr3 expression by tetracycline in a HeLa cell line was demonstrated to co-localize with actin at the edge of spreading cells [30]. The RPEL domain of Phactr4 does not control its nuclear localization [31]. Actin binding by the three Phactr1 C-terminal RPEL motifs is required for Phactr1 cytoplasmic localization [34].

To determine whether the replacement of Arg 511 by His affects the subcellular distribution of PHACTR2, we transfected Hek293 and HeLa cell lines with lentiviral constructs containing either the normal coding sequence or PHACTR2 with 511 His fused to myc tags at the carboxy terminus. The subcellular distribution was visualized using the tagged myc antibody by confocal microscopy. We found that the substitution did not affect the localization of PHACTR2 to the cytoplasm (Figure 2). Additionally, PHACTR2 did not seem to co-localize with the cellular filamentous actin, which was visualized by fluorescent phalloidin (Figure 2, zoom). Similarly, PHACTR1 was demonstrated as not co-localizing with filamentous actin [12]. When overexpressed, each Phactr family member led to a change in the cell shape and the formation of cell protrusions of a variable length and direction, highlighting a role in regulating actin cytoskeleton dynamics [31,36]. We did not observe any such changes in the cell shapes; thus, we assumed we did not overexpress either form of PHACTR2.

### 2.5. Globular Actin Pool in Patient’s Fibroblasts

The interaction of Phactr proteins with cytoplasmic actin suggests that this protein family may regulate cytoskeletal dynamics. The PREL domain of Phactr4 and Phactr1 facilitates the competitive binding of monomeric actin and PP1 [31,34].

To verify whether PHACTR2 participates in actin dynamics and whether the replacement of Arg 511 by His may affect actin dynamics, we first asked whether the binding of the variant PHACTR2 to the actin was reduced, as expected by the structural analysis (Figure 1D) affecting the ratio between globular and filamentous actin. Thus, we studied the glomerular and filamentous actin ratio in the patient’s fibroblasts. The globular and filamentous actin can be separated by lysing with an actin stabilization buffer, followed by centrifugation, which leaves the filamentous actin in the pellet and the globular actin together with other cytoplasmic proteins in the supernatant. The presence of GAPDH reflects the separation efficiency only in the supernatant [37]. Indeed, we found less globular actin in the patient’s fibroblasts compared to the control (Figure 3). This result also agrees with the report that the overexpression of Phactr4 led to increased levels of actin monomers in cells [31].

### 2.6. Kinetics of Actin Repolymerization in Patient’s Fibroblasts

To further study the effect of the change in Arg 511 to His in PHACTR2 on actin dynamics, we followed the kinetics of actin repolymerization upon depolymerized filamentous actin. We treated the fibroblast cells with cytochalasin D (CD), causing actin depolymerization [38]. After removing the drug, we measured the kinetics of the reformation of actin filaments. One hour after treatment with 30 µM CD, the actin filaments of the patient cells were almost completely disrupted. In contrast, the control cells still presented a few intact filaments (Figure 4A zero time point), suggesting that the actin filaments of the patient cells were more accessible to the depolymerization process than the filaments of the control cells. In addition to the actin depolymerization, patient cells lost their long, stretched morphology and presented a rounded shape. In contrast, the morphology of the control cells was significantly less affected. After the washout of the drug and further incubation at 37 °C, the reformation of the actin filaments was markedly different between the patient and control cells at 15 min, 30 min, and 60 min of recovery. At 60 min, the recovery of the actin filaments appeared complete in the control group, whereas the recovery in the patient group was still affected. A similar picture for the cell shape was obtained, with a marked statistical difference between the patient and control cells at 15 min, 30 min, and 60 min of recovery (Figure 4B). We quantified the depolymerized actin filaments by counting the dots representing unpolymerized actin using ImageJ software. We found a significantly lower amount of depolymerized actin filaments in the control group than in the patient group at 0 min and at all time points of recovery (Figure 4C). Assuming the PHACTR2 variant is the only variant to be found that causes the disease, these results may suggest that PHACTR2 is essential for actin polymerization.

### 2.7. Migration Assay of Control and Patient Fibroblast Cells (Wound Healing Assay)

Many PHACTR family members seem to be required for cell motility and proliferation. PHACTR1 knockdown reduced mouse brain capillary endothelial cell migration and promoted the expression of apoptosis-associated proteins [39]. On the other hand, the overexpression of PHACTR4 inhibited hepatic cell carcinoma (HCC) cell proliferation, colony formation, migration, and invasion and resulted in significant cycle arrest [40]. PHACTR4 was also demonstrated to modulate integrin signaling and cofilin activity to coordinate the forces that drive enteric neural crest cell (ENCC) migration in the mammalian embryo [33]. The expression of PHACTR3 in HeLa cells stimulated cell spreading and motility, dependent on the RPEL domain [30]. Thus, we verified proliferation and cell motility.

We compared cell motility by the wound healing assay and showed differences between the migration and proliferation of the control and patient fibroblast cells at 0, 24, and 48 h in the experiment (Figure 5A). The control fibroblasts bridged the gap significantly faster than the patient fibroblasts. The first contact of fibroblasts from both sides of the gap was considerably shorter in the control than in the patient fibroblasts (Figure 5B). The control fibroblasts covered the gap interval entirely in ~30 h, whereas the patient cells did not close the gap even after 67 h (the end of the experiment) (Figure 5C). Moreover, the migration direction was disturbed in the patient compared to the control fibroblasts (video in the Appendix A).

## 3. Discussion

Here, we demonstrate a possible association between a missense variation in *PHACTR2* and a clinical manifestation of DCM and LVNC in a human patient for the first time. The early and severe presentation of our patient agrees with a study of 350 DCM patients, which found that the average patient age was significantly younger in the AR form of DCM compared to the AD form, with more rapid progression to death or heart transplant [5]. We suggest that the missense variation possibly disrupts the binding of PHACTR2 to globular actin and affects cellular actin dynamics, proliferation, and the movement of the patient’s cells. The variation was segregated as expected in the family members and was not detected in the 1714 chromosomes of the Bedouin controls. The clinical findings are restricted to the heart, although the gene is ubiquitously expressed (GTEx portal), and a knock-out mouse model does not present any significant phenotype (mousephenotype organization). Thus, the variation may have a specific effect on the heart muscle, which differs from the complete absence of the gene product. The patient’s lack of a skeletal muscle phenotype may partly result from the twice lower expression of *PHACTR2* in the skeletal muscle (GTEx).

The missense variation in *PHACTR2* does not affect its sub-cellular distribution, which remains in the cytoplasm in agreement with previous studies [31]. The variation appears to affect the binding to globular actin, as predicted by the structural analysis and demonstrated by the decrease in the globular actin pool in the patient fibroblasts. Indeed, we did not note an association between the PHACTR2 protein and fibrilar actin (Figure 2). Competitive binding between globular actin and PP1 was shown for both PHACTR4 [31] and PHACTR1 [34]. Therefore, it was suggested that the PHACTR–PP1 complex modulates cofilin or myosin’s phosphorylation status, thus regulating actin cytoskeleton dynamics [31,33,34]. We suggest that PHACTR2 may act similarly. The disruption of actin-binding probably perturbs the polymerization to filamentous actin and the migration of the patient’s fibroblasts which depend on the kinetics of actin [30,33]. Fibrillary actin maintains cell morphology, spatial structure, cell adhesion, and cell transport [41]. We have not observed the formation of actomyosin foci or thickened actin stress fibers, as reported for a non-actin binding PHACTR1 mutant in NIH3T3 fibroblasts [34]. In agreement with our finding of the defective repolymerization of actin, PHACTR1 gene silencing significantly reduced the polymerization in HUVEC cells treated with VEGF-A165 [42].

The defective migration and proliferation of the patient fibroblasts agree with the findings of other PHACTR members: *PHACTR4* is required for directional migration [33]; the expression of *PHACTR3* markedly enhanced cell motility [30]. However, both PHACTR4 and PHACTR3 localized to the cell membranes at the leading edges of the cells [30,33], which we did not observe for either normal or variant PHACTR2. PHACTR2 may act similarly to PHACTR1, which, when knocked down, reduces mouse brain capillary endothelial cell line proliferation and migration in the scratch assay by downregulating the expressions of migration-associated proteins (matrix metalloproteinase MMP-2 and MMP-9) and upregulating apoptosis-associated proteins (Bax, Bcl-2, cleaved caspase-3, and caspase-3) [39]. Our study is the first demonstration that *PHACTR2*, the lesser-studied *PHACTR* member, may have a role in cellular migration and proliferation.

Our study is limited by identifying this variation in only one patient in terms of the association between the *PHACTR2* function and the phenotypic presentation in the heart; additionally, the absence of the gene does not cause a mouse phenotype. The mouse and human phenotypes caused by the inactivity of genes may differ [43]. Moreover, the phenotype in the patient may be caused by a combination of the heterozygous TNNT2/TTN missense variations (Appendix A) and the homozygous PHACTR2 variant. However, here we present a study demonstrating that actin dynamics are affected in the patient’s fibroblasts: the patient’s fibroblasts have a reduced glomerular to filamentous actin ratio, and the actin filaments of the patient cells were more accessible to the depolymerization process by CD than the filaments of the control cells. The kinetics of repolymerization were slower, and measuring cell motility by the wound healing assay showed that the patient cells did not close the gap, and their migration direction was disturbed.

*PHACTR2* can be linked to the cardiac phenotype since it was reported to interact with sarcomeric cardiac actin [13]. Indeed, mutations in ACTC1, encoding cardiac actin, were present in 2% of the patients reported in an extensive study on 840 LVNC and DCM cases compared to 125,748 gnomAD population controls [8]. Furthermore, in agreement with the pediatric presentation of our patient, ACTC1 non-truncating variants, acting in a dominant negative mechanism, were enriched in children [8,44]. The specific representation of our patient’s heart phenotype could be attributed to a disturbance caused by the replacement of Arg 511 by His in the interaction of PHACTR2 with PP1. PP1 is a major eukaryotic serine/threonine protein phosphatase that regulates diverse cellular processes, including muscle contractions and actin cytoskeleton organization [45]. It is not freely available within the cardiac cells, but rather a competition of >150 regulatory subunits, which form a holo-complex with the PP1 catalytic subunits. These regulatory subunits determine the subcellular localization and substrate specificity of the different PP1 isoform catalytic subunits [46]. Additional studies examining whether the changes in actin dynamics could lead to alterations in the actin-binding protein expression (e.g., cofilin, filamins, troponins, alfa actinin, etc.) would be interesting.

Thus, *PHACTR2* may join other regulators of actin dynamics that have been demonstrated to cause human genetic diseases. Our study is the first to show a variation in *PHACTR2* in humans and its effect on the perturbation of actin dynamics, which can cause DCM.

## 4. Patient and Methods

### 4.1. Patient

The study was approved by the Soroka University Medical Center Institutional Review Board. All subjects involved in the study obtained informed consent prior to participation. Medical records were carefully reviewed, and details of somatic growth, psychomotor development, clinical course, hospitalizations, and laboratory results were obtained. Parents and siblings were interviewed and underwent a complete physical examination that particularly focused on cardiac and neuromuscular findings. A cardiac evaluation was performed for all participants, including echocardiography and electrocardiography [43].

### 4.2. Genetic Analysis

Genomic DNA (gDNA) was extracted from the blood of all individuals of generations I + II (Figure 1A). Fibroblast cells were established from a skin biopsy of patient II-3.

The gDNA of Patient II-3 was submitted to Otogenetics Corporation (Norcross, GA, USA) for exome capture and sequencing. gDNA was subjected to agarose gel and OD ratio tests to confirm the purity and concentration prior to Covaris (Covaris, Inc., Woburn, MA, USA) fragmentation. Fragmented gDNAs were tested for their size distribution and concentration using an Agilent Tapestation and Nanodrop. Illumina libraries were made from qualified fragmented gDNA using NEBNext reagents (New England Biolabs, Ipswich, MA, USA, catalog# E6040), and the resulting libraries were subjected to exome enrichment using NimbleGen SeqCap EZ Human Exome Library v2.0 (Roche NimbleGen, Inc., Madison, WI, USA, catalog# 05860482001) following the manufacturer’s instructions. Enriched libraries were tested for enrichment by qPCR and for size distribution and concentration by an Agilent Bioanalyzer 2100. The samples were then sequenced on an Illumina HiSeq2000, which generated paired-end reads of 90 or 100 nucleotides. The average depth of the sequence was ×30. Data were analyzed for quality, exome coverage, and exome-wide SNP/InDel using the platform provided by DNAnexus (DNAnexus, Inc, Mountain View, CA, USA).

Fabric Genomics (https://www.fabricgenomics.com/, accessed on 29 July 2018), Franklin by Genoox (https://franklin.genoox.com/, accessed on 13 December 2022), ClinVar, and ACMG criteria were used to evaluate the pathogenicity of the variants. The damage score of Omicia used six prediction algorithms: MutationTaster, Polyphen-2—HDIV, SIFT, phyloP—Vertebrate, phyloP—Placental, VVP (VAAST Variant Prioritization), and CADD (Combined Annotation Dependent Depletion). Franklin’s aggregated prediction algorithm used the predictions tools—SIFT, FATHMM (functional analysis through hidden Markov models), DANN (deleterious annotation of genetic variants using neural networks), REVEL (rare exome variant ensemble learner), MutationAssessor, PolyPhen-2, MutationTaster, PrimateAI, BayesDel, and GERP (genomic evolutionary rate profiling). The databases used to exclude variants based on their frequency in the general population were the public databases (gnomAD browser, 1000 genomes, EVS, and ExAC). Our in-house Bedouin exome sequence database, created while looking for mutations in this population, excluding heart patients, included 780 sequences. An additional database for healthy Bedouins [27] and more than 1000 exomes of Qatari genomes for most of the Bedouin population [28] were used to exclude frequent variants. The GeniePool database, including thousands of NCBI’s sequence read archive (SRA) exomes, helped us narrow down the list of the heterozygous variations. We used OMIM and GeneCards for the clinical presentations of the associated genes and genotype-tissue expression (GTEx) to look for the cardiac gene expression.

### 4.3. Variation Verification

The PCR amplification of exon 8 for the PHACTR2 gene (Gene ID: 20956; NM_014721.3) was performed using primers forward: 5′ATACGCCGGAGGGATACTCT3′ and reverse 5′TTCACAATGACCCCAGTTAGG3′ (annealing temperature 62 °C). Direct sequencing of the PCR products was performed as detailed [43].

### 4.4. Protein Structure

The predictions of the domains, repeats, motifs, and features within Ape’s (Pongo abellii) PHACTR2 protein isoform 4 were retrieved from the SMART database (simple modular architecture research tool, http://smart.embl-heidelberg.de/, accessed on 20 February 2018). The 3D structure modeling of the PHACTR2 protein was created using the Swiss model server. The amino acid sequence of human PHACTR2 was retrieved from the NCBI protein database (NP_055536.2). Since the residues around the variation site were conserved, we used the crystal structure of mouse PHACTR1 with actin (PDB structure 4B1Z) for the structural analysis. Color modifications on the protein were changed using PyMOL software.

### 4.5. Lentivirus Preparation and Infection

Lentiviral packaging of the desired transgenes was performed using standard techniques. Hek293T cells were plated in 6-well plates and allowed to grow to ~50% confluency. They were transfected using the jetPRIME transfection protocol (Polyplus, #101000027, https://www.polyplus-transfection.com/products/jetprime/, accessed on 5 July 2018), with 6 µg psPAX2 (Addgene plasmid, #12260), 4 µg pMD2.G (Addgene plasmid, #12259) and 10 µg of the transgene plasmid containing lentiviral coding sequences of the WT and Mut PHACTR2 gene fused to myc tag (pLV[Exp]-Puro-CMV>hPHACTR2[NM_014721.2]/Myc, VectorBuilder). The media was changed after 4 h. Following 48 h of incubation, the media was aspirated, centrifuged at 2000rcf for 5 min at 25 °C to remove the cell debris, and filtered through a 0.45 µm PVDF membrane millex syringe filter (Merck-Millipore, MA, USA).

For viral infections, cells were plated in 12-well plates and allowed to grow to ~40% confluency. Thawed virus-containing media, supplemented with 8 µg/mL polybrene, was added to the cells, and infection was performed by ‘spinoculation’ at 1200× *g*, 32 °C for 90 min. Virus-containing media was aspirated and replaced with fresh media, and cells were allowed to recover for 24 h. When appropriate, the infected cells were selected with 1–10 µg/mL puromycin until no living cells were detected in the control wells.

### 4.6. Immunofluorescence Studies

Hek293 and HeLa cells were grown in an 8-well slide (ibidi 80826) in DMEM with 10% FBS (Biological Industries, Beit Haemek, Israel). Cells were fixed with 1:1 ethanol methanol solution at −20 °C and placed in 1% bovine serum albumin (BSA) (Sigma-Aldrich) in PBS for 1 h to block non-specific binding. The samples were incubated at room temperature for two hours with an anti-myc (present from Prof. Noah Isakov) antibody. Cells were stained with Alexa-Fluor 546-conjugated phalloidin (A22283; Life Technologies) and Cy2 conjugated anti-c-myc (ZRB1003; Sigma-Aldrich) for 1 h at room temperature, followed by three washes with PBS ×1 (5 min for each). Nuclei were visualized by adding Hoechst 33342 (H3570; Life Technologies). Images were acquired with the confocal FluoView 1000 fluorescence microscope (OLYMPUS), with a 60× objective and the manufacturer’s software FV1000.

### 4.7. Kinetics of Actin Re-Polymerization

The treatment of Fibroblast cells with cytochalasin D and staining were performed as detailed for the treatment with cytochalasin B [47]. Cells were fixed with 4% formaldehyde solution, incubated with 0.1% Triton X-100, and blocked with 1% bovine serum albumin (BSA) (Sigma-Aldrich). Images were acquired using laser scanning microscopy with super-resolution (Zeiss LSM880 Airyscan), with a 40/20× objective, and with the manufacturer’s software Zen Lite. Images were processed using the Image J software.

### 4.8. Fractionation of Globular and Filamentous Actin

Western blot was performed as detailed [47]. The blot was incubated with the primary anti-beta actin antibody (1:5000 in 5% milk) (ab8227; Abcam) against G and F-actin, and the GAPDH antibody (1:2000 in 5% milk) (Millipore MAB374) against GAPDH. The secondary antibody was a horseradish peroxide conjugated anti-mouse IgG (cat. no. 1706516, Bio-Rad, USA). The membrane was visualized using EZ-ECL (20-500-120; Biological Industries) on the Fusion instrument and analyzed using the Fusion software (A2S). For determining the G/F actin ratio, cells were washed once in ice-cold PBS and lysed with an actin stabilization buffer (0.1 M PIPES, pH 6.9, 30% glycerol, 5% DMSO, 1 mM MgSO4, 1 mM EGTA, 1% TX-100, 1 mM ATP, and protease inhibitor) on ice for 10 min. Cells were dislodged by scraping and centrifuged at 4 °C for 75 min at 16,000× *g*. The supernatant (G-actin and GAPDH) and the pellet (F-actin) fractions were resolved on 12% SDS-PAGE gels and analyzed by Western blot, as described above. The GAPDH was used to assure the separation’s success as it appears only in the supernatant fraction and quantifies the loaded lysate. We used the protocol as detailed [37].

### 4.9. Cell Migration Assay

Analysis was performed as detailed [43]. Fibroblasts of identical passages (11 to 12) of patient II3 and the control fibroblasts (derived from the foreskin) were plated with culture inserts of 500 μm. When the cell density reached ~90%, the inserts were removed, thus creating a ~500 μm-wide gap within the culture. The movement toward closing the gap was monitored using a time-lapse every 20 min for 67 h by differential interference contrast (DIC) images in a live-cell FluoView 1000 fluorescence microscope (OLYMPUS), with a 20× objective and with the manufacturer’s software FV1000. The contact time was defined as the time that passed from the insert removal until the first cell contact from both sides of the gap. Coverage was calculated by dividing each microscope figure (covering an area of 1260 µm^2^) into nine squares and measuring the area covered by the cells in each square. The results represent the means of all squares. The coverage was calculated in an interval of 5 h.

### 4.10. Statistical Analysis

Values are expressed as means ± SD. A comparison of the cellular morphology and the depolymerized actin count at different time points following cytochalasin D application was made using a Kruskal Walis non-parametric test with Dunn’s multiple comparisons post-test. Measuring the depolymerized actin count was performed by ImageJ software version 1.53i. The criterion for significance was set at *p* < 0.05. Unless otherwise stated, *p*-values are displayed graphically as follows: * *p* ≤ 0.005, ** *p* ≤ 0.001, *** *p* ≤ 0.0005.

## Figures and Tables

**Figure 1 ijms-24-01388-f001:**
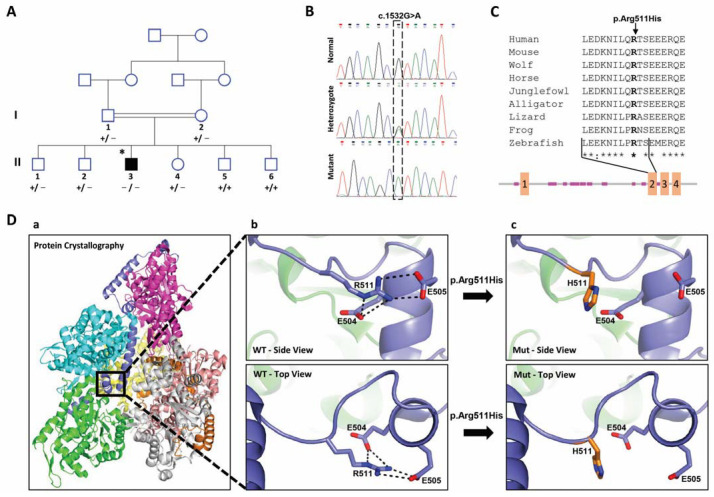
Family pedigree, Sanger sequencing of the mutant locus, evolutionary conservation, domain location, and 3D structure of the protein. (**A**) Pedigree and segregation of the variation in the *PHACTR2* gene. The presence of the *PHACTR2* variation in the family was determined by PCR amplification of the region that contains the variation. +/+ normal, −/+ heterozygous, −/− homozygous for the mutation. * Indicates an individual for whom whole-exome sequencing (WES) was conducted. Individuals in the pedigree who were not clinically assessed appear without numbers underneath. (**B**) Example chromatograms of Sanger sequences from individuals homozygous for the variation, heterozygous, or normal. (**C**) The upper part shows the variation’s evolutionary conservation in the protein region. The lower part consists of a diagram that shows the predicted domains, repeats, motifs, and features within Ape’s (Pongo abellii) PHACTR2 protein isoform 4 by the SMART database, thus presenting the second RPEL domain that contains the variation. Pink rectangles represent low-complexity domains, while orange represents the four numbered RPEL domains. (**D**) Modeling the 3D structure of the PHACTR2 protein using the Swiss model server. (**a**) Represents the overall 3D structure of the protein based on the mouse PDB structure 4B1Z. (**b**) Magnification of the site of the affected amino acid, Arg (the right helix part of the figure), and its hydrogen bond to the Ser amino acid. (**c**) Replacement of the amino acid Arg to His due to the variation, causing the loss of hydrogen bonds.

**Figure 2 ijms-24-01388-f002:**
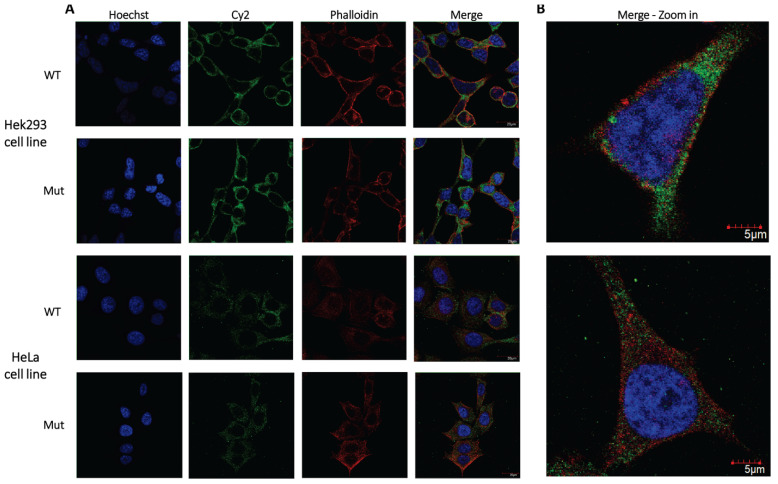
Subcellular localization of the normal and variant PHACTR2 proteins and co-localization of PHACTR2 and actin. (**A**) Hek293 and HeLa cell lines were infected with Lentiviruses containing the normal (WT) or His511 (Mut) PHACTR2 coding sequence fused to myc tags. The subcellular localization of the PHACTR2 construct was visualized by confocal microscopy. PHACTR2 protein is marked by Cy2 in green, F-actin by phalloidin-Alexa Fluor 546 in red, and nuclear staining by Hoechst 33342 in blue. (**B**) Merged magnification of WT PHACTR2 staining for each cell line, respectively, demonstrates that PHACTR2 does not co-localize with F-actin.

**Figure 3 ijms-24-01388-f003:**
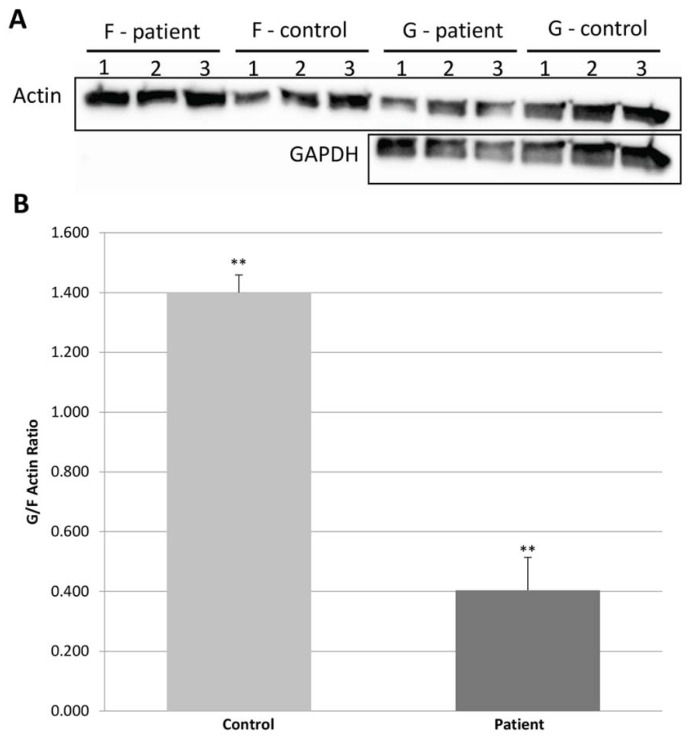
PHACTR2 gene variation significantly decreases the ratio between globular (G) and filamentous (F) actin. (**A**) Representative Western blots showing the G-actin content relative to F-actin in triplicates of patient fibroblast cells compared to control fibroblast cells. GAPDH was used as an internal loading control for the G-actin fraction. (**B**) Western blot results were used to measure the G-actin to F-actin ratio using Fusion software (A2S) while comparing the patient’s fibroblast cells to control fibroblast cells. The results represent three independent experiments; each experiment was verified by triplicate Western analyses. Error bars are SD. ** *p*  ≤  0.001.

**Figure 4 ijms-24-01388-f004:**
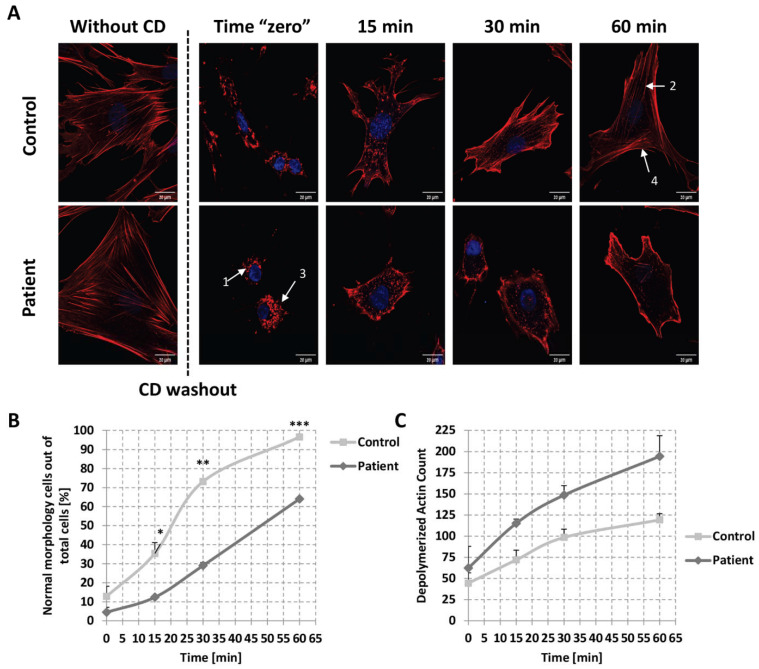
Effect of changing Arg 511 to His on the kinetics of the repolymerization of actin and cell morphology in patient and control fibroblasts. (**A**) The recovery of actin filaments in control and patient fibroblasts after disruption with cytochalasin D (CD) and at various time points after removing the drug was visualized by confocal microscopy using fluorescent phalloidin. The arrows indicate (1) completely disassembled actin; (2) repolymerized actin; (3) round cell morphology, which may represent detachment from the plate; and (4) normal cellular morphology. (**B**,**C**) Quantification of the depolymerized actin filaments and the cell morphology, respectively. Results represent n = 6 from three independent cultures. Error bars are SD. * *p*  ≤  0.005, ** *p*  ≤  0.001, *** *p*  ≤  0.0005.

**Figure 5 ijms-24-01388-f005:**
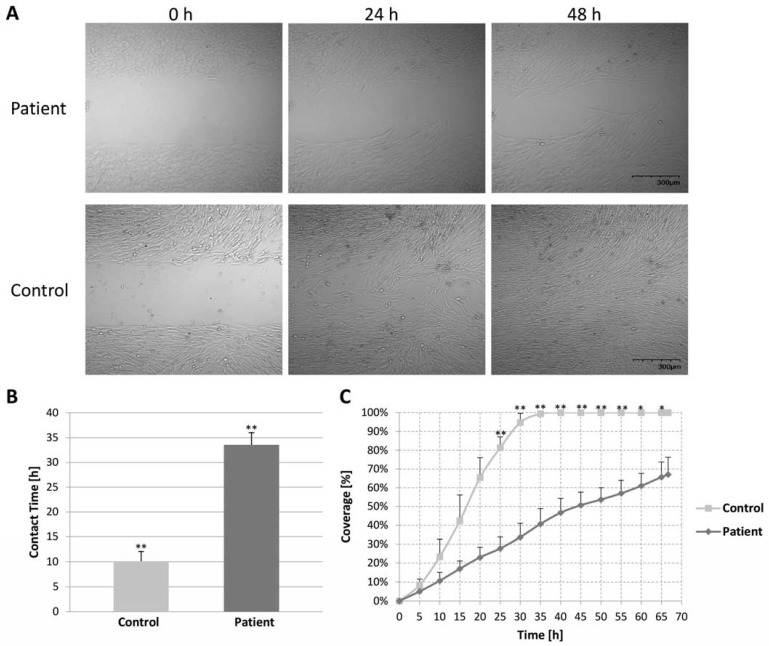
Wound healing assay assessing control and patient fibroblasts’ migration and plate coverage. Fibroblast movement was monitored using a time-lapse every 20 min for 67 h by a live-cell microscope. (**A**) Images of the control and patient’s fibroblasts were taken after removing the ibidi culture-insert 2 wells at 0, 24, and 48 h. (**B**) Migration time over the gap was determined by measuring the contact time of the cells from each side of the gap. The patient’s fibroblast cells noticed a significant increase in contact time. (**C**) Coverage was calculated by dividing each microscope figure into nine squares and measuring the area covered by the cells in each square. The results represent the means of all squares. The coverage was calculated at intervals of 5 hrs. There was a significant time difference in which the cells covered the plate. All results are representative of three independent experiments. Error bars are SD. * *p* ≤  0.005, ** *p* ≤  0.001.

**Table 1 ijms-24-01388-t001:** Clinical evaluation of the DCM patient—II-3.

Age at Onset and Clinical Situation	Age at Follow-Up and Clinical Situation (p/s)	Echo at Follow-Up, 10 Wks	Age at Follow-Up and Clinical Situation (p/s)	Echo at Follow-Up, Teens	CMRI in Teens Cine-SSFP	Repeated Cardiac Holter Monitoring
Prenatal diagnosis at 32 wks Cardiomegaly Increased cardiac/thoracic Circumference ratio: 0.7 (normal: 0.45–0.5)	10 wks Congestive heart failure Dyspnea Tachypnea Tachycardia Eating difficulty	LVEDD 32 mm z-score- +3.84 LVESD 25 mm z-score- +4.89 EF- 40%	Teens Constant fatigue Exercise intolerance	LVEDD 52 mm z-score- +1.67 LVESD 43 mm z-score- +3.59 EF- 37–40%	LVEDV 101 mL/m^2^ LVESV 64 mL/m^2^ EF- 44% Normal origin of coronary artery Non-compaction LV	22 h Normal sinus rhythm

The annotation of the patient is according to Figure 1A. LVESD—left ventricular end-systolic diameter; LVEDD—left ventricular end-diastolic diameter; EF—ejection fraction; LVESV—left ventricular end-systolic volume; LVEDV—left ventricular end-diastolic volume.

## Data Availability

The data will be made available upon request.

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
