# Peer review of "A Missense Variation in PHACTR2 Associates with Impaired Actin Dynamics, Dilated Cardiomyopathy, and Left Ventricular Non-Compaction in Humans"

_ijms, 2023, doi:10.3390/ijms24021388_

Round 1
Reviewer 1 Report
ijms-2055587
This paper entitled “
A missense variation in PHACTR2 associates with impaired ac-2 tin dynamics, dilated cardiomyopathy, and left ventricular non-3 compaction in humans” used whole-exome sequencing to investigate a possible genetic cause of DCM with LVNC in a consanguineous child, a homozygous nucleotide change c.1532G>A causing p.Arg511His in PHACTR2 was found. The missense change can affect the binding of PHACTR2 to actin by eliminating hydrogen bonds between them. The amino acid change does not change PHACTR2 localization to the cytoplasm. The patient's fibroblasts show a decreased globular to fibrillary actin ratio compared to control fibroblasts. Re-polymerization of fibrillary actin after treatment with cytochalasin D, which disrupts the actin filaments, is slower in the patient's fibroblasts. Finally, the patient's fibroblasts bridged a scar gap slower than the control fibroblasts because of slower and indirect movement.
The authors claims that this is the first report of a human variation in a PHACTR family member. The knock-out mouse model presents no significant phenotype. They think the data underscore the importance of PHACTR2 in regulating the monomeric actin pool, the kinetics of actin polymerization, and cell movement, emphasizing the importance of actin regulation for the normal function of the human heart.
The study is interesting but there are deficiencies.
Critiques:
1. WES was done only in the proband. It is better to have WES done in the proband and the parents. The identification of variations in PHACTR2 may be combined with variations in other genes as well. This can be excluded with the WES analysis on the parents gDNA.
2. “ No human phenotype was yet associated with any of the PHACTR family members. line 73-73” Is not accurate. Through OMIM, it is found that the phenotype of PHACTR1 is Developmental and epileptic encephalopathy 70”
3. figure1 D, mouse PHACTR1protein 3D structure was used to predict the impact of the variations of PHACTR2 in human. However, the identity of human and mouse on PHACTR1 is as low as 30%; In contrast mouse PHACTR2 has higher identity with human PHACTR2 (60%). Therefore, it would be better to use mouse PHACTR2.
4. The authors identified three missense heterozygous variations with less than 0.5% allele frequencies in the public databases (gnomAD browser, 1000 Ge-128 nomes, ExAC and EVS) in these genes in patient II-3. Taking 0.5% as the cut-off is too high (the standard is too low.) Usually less than 0.1% is more acceptable standard.
Author Response
Response to Reviewer 1 Comments
Point 1: WES was done only in the proband. It is better to have WES done in the proband and the parents. The identification of variations in PHACTR2 may be combined with variations in other genes as well. This can be excluded with the WES analysis on the parents gDNA.
Response 1: We hypothesized that pathological variations are causing the patient's disease and, indeed, these variations would be inherited from the parents. Accordingly, and instead of trio WES we verified each of the suspected heterozygous/homozygous variations that met the exclusion criteria identified in the patient, in his parents and the healthy siblings. The segregation was confirmed by PCR amplification of the DNA containing the variation, followed by Sanger sequencing, thus enabling the exclusion of variations based on the gDNA of the parents. Please see "Identification of the PHACTR2 variation" section in the Results and Supplementary tables 3, 4, and 5 for more details. Furthermore, the possibility of combined variations leading to the disease's phenotype was discussed in the "Discussion" section under the limitation paragraph, lines 385-387.
Point 2: "No human phenotype was yet associated with any of the PHACTR family members. line 73-73" Is not accurate. Through OMIM, it is found that the phenotype of PHACTR1 is Developmental and epileptic encephalopathy 70".
Response 2: Thank you for this important correction; we changed this sentence in the abstract: "This is the first report of a human variation in this PHACTR family member.". We deleted the sentence in lines 76-77 "No human phenotype was yet associated with any of the PHACTR family members." and replaced it with a corrected one, “PHACTR1 is the only PHACTR family member associated with a human phenotype. Three de novo mutations in the PHACTR1 gene were found to associate with developmental and epileptic encephalopathy (OMIM *608723). “.
Point 3: figure1 D, mouse PHACTR1protein 3D structure was used to predict the impact of the variations of PHACTR2 in human. However, the identity of human and mouse on PHACTR1 is as low as 30%; In contrast mouse PHACTR2 has higher identity with human PHACTR2 (60%). Therefore, it would be better to use mouse PHACTR2.
Response 3: We agree with the reviewer's point that it would have been better to verify the effect of the variation on the PHACTR2 structure and look for its structure in the protein data bank. Unfortunately, there is no experimental structure for PHACTR2 but only for PHACTR1. The confidence of the Alpha fold and Robetta predicted structures of PHACTR2 is low, and they are less reliable than the experimental structure of PHACTR1. Since the variant's location is conserved between PHACTR1 and PHACTR2 around the site, we feel confident about using the structure of mouse PHACTR1. We added a sentence to the manuscript, lines 179-182. "We verified the effect of the candidate variation on the experimental crystal structure of mouse PHACTR1, which is conserved between PHACTR1 and PHACTR2 at the variant location site since there is no experimental structure of PHACTR2."
Point 4: The authors identified three missense heterozygous variations with less than 0.5% allele frequencies in the public databases (gnomAD browser, 1000 Ge-128 nomes, ExAC and EVS) in these genes in patient II-3. Taking 0.5% as the cut-off is too high (the standard is too low.) Usually less than 0.1% is more acceptable standard.
Response 4: Indeed, 0.1% is usually the accepted standard to exclude variants by allele frequency in the public database. We used 0.5% allele frequency as a cut-off, so we do not miss any possible pathological variation for two reasons-
- We worked with a Bedouin kindred, characterized by a high frequency of consanguineous marriages; thus, their allelic frequency may not be concurrent with the public databases (gnomAD browser, 1000 Genomes, ExAC and EVS).
- We looked for the possibility that the patient's phenotype may be caused by a combination of heterozygous variations in genes reported having mutations causing familial and non-familial DCM (Supplementary Table 2) along with the homozygous PHACTR2 variant. Therefore, to avoid missing any variation, we used a higher allelic frequency cut-off for the identification of combined variations that might lead to the disease's phenotype.
- The paper was edited by MDPI language editing services. English editing certificate number 55743.
Please see the attachment of the edited paper.

Reviewer 2 Report
The paper needs language editing by native speaker
Author Response
Response to Reviewer 2 Comments
Point 1: The paper needs language editing by native speaker
Response 1: The paper was edited by MDPI language editing services. A native English speaker checked the grammar, spelling, punctuation, and phrasing of the paper to improve its readability.
English editing certificate number is 55743.
Please see the attachment of the edited paper file.

Reviewer 3 Report
This article by P. Majdalani et al., demonstrated a missense variation in PHACTR2 gene contributing to alteration in actin dynamics and possible association to dilated cardiomyopathy and left ventricular non compaction in a patient from Bedouin family. Further, using in vitro experiments, they have demonstrated that this variation in the PHACTR2 gene results in altered actin dynamics and migration of cells etc.
This is a very good study and potentially very important observation highlighting the association of a previously uncharacterized VUS in PHACTR2 gene possibly contributing to DCM development. Authors should be commended for a such a clear cut rationale and the methodology employed. I would definitely consider accepting this manuscript for possible publication in IJMS, pending these small minor revisions.
1. Additional studies examining whether the changes in actin dynamics would lead to alteration in acting binding protein expression (e.g. cofilin, filamins, troponins, alfa actinin etc.), would be interesting. These can be potentially studied in commercially available primary human cardiomyocytes. The changes observed if any, would be very valuable and relevant for the cardiac functional changes observed in the patient.
2. I am not sure why authors have mentioned in the figure 3 legend that the GAPDH was used as internal loading control for only G- actin fraction. Why can’t this be a loading control for the F-fraction? It is also not clear from the western blot method, what antibodies have been used for the detection of respective G and F forms of actin? Please elaborate the method so that the readers can understand how this is done.
Author Response
Response to Reviewer 3 Comments
Point 1: Additional studies examining whether the changes in actin dynamics would lead to alteration in acting binding protein expression (e.g. cofilin, filamins, troponins, alfa actinin etc.), would be interesting. These can be potentially studied in commercially available primary human cardiomyocytes. The changes observed if any, would be very valuable and relevant for the cardiac functional changes observed in the patient.
Response 1: Thank you for the valuable study suggestion. Indeed, looking for actin-binding protein expression alterations would contribute to a better understanding of the functional changes observed in the patient's cardiac tissue. Unfortunately, these experiments are expensive, demanding, time-consuming, and out of our ability. They will require to order the specific study cell line, having the cells infected with the Lentiviral containing WT/Mutated protein, and afterward proceeding with the expression analysis, thus taking much more time than the time required for this review. We have added this excellent suggestion of the reviewer to the discussion, lines 408-410:" Additional studies examining whether the changes in actin dynamics would lead to alteration in actin-binding protein expression (e.g., cofilin, filamins, troponins, alfa actinin, etc.) would be interesting."
Point 2: I am not sure why authors have mentioned in the figure 3 legend that the GAPDH was used as internal loading control for only G- actin fraction. Why can't this be a loading control for the F-fraction? It is also not clear from the western blot method, what antibodies have been used for the detection of respective G and F forms of actin? Please elaborate the method so that the readers can understand how this is done.
Response 2: We thank the reviewer for this valuable comment, realizing that this part was not clearly explained. We added now lines 257-261: "The globular and filamentous actin can be separated by lysing with an actin stabilization buffer, followed by centrifugation, which leaves the filamentous actin in the pellet and the globular actin together with other cytoplasmic proteins in the supernatant. The presence of GAPDH reflects the separation efficiency only in the supernatant [47]."
In the methods, we changed the title from "Western blot analysis" to "Fractionation of globular and filamentous actin" (line 503). We detailed the buffer, centrifugation conditions, and antibody used in lines 503-517. We wrote: "The blot was incubated with the primary anti-beta actin antibody (1:5000 in 5% milk) (ab8227; Abcam) against G and F-actin, and GAPDH antibody (1:2000 in 5% milk) (Millipore MAB374) against GAPDH. The secondary antibody was horseradish peroxide conjugated anti-mouse IgG (cat. no. 1706516, Bio-Rad, USA)." and continued, "For determining the G/F actin ratio, cells were washed once in ice-cold PBS and lysed with actin stabilization buffer (0.1 M PIPES, pH 6.9, 30% glycerol, 5% DMSO, 1 mM MgSO4, 1 mM EGTA, 1% TX-100, 1 mM ATP, and protease inhibitor) on ice for 10 min. Cells were dislodged by scraping and centrifuged at 4◦C for 75 min at 16,000 × g. The supernatant (G-actin and GAPDH) and the pellet (F-actin) fractions were resolved on 12% SDS-PAGE gels and analyzed by Western blot, as described above. The GAPDH was used to assure the separation's success as it appears only in the supernatant fraction and quantifies the loaded lysate.".
- The paper was edited by MDPI language editing services. English editing certificate number is 55743.
Please see the attachment of the edited paper file.

Round 2
Reviewer 1 Report
Response to Reviewer 1 Comments
Point 1: WES was done only in the proband. It is better to have WES done in the proband and the parents. The identification of variations in PHACTR2 may be combined with variations in other genes as well. This can be excluded with the WES analysis on the parents gDNA.
Response 1: We hypothesized that pathological variations are causing the patient's disease and, indeed, these variations would be inherited from the parents. Accordingly, and instead of trio WES we verified each of the suspected heterozygous/homozygous variations that met the exclusion criteria identified in the patient, in his parents and the healthy siblings. The segregation was confirmed by PCR amplification of the DNA containing the variation, followed by Sanger sequencing, thus enabling the exclusion of variations based on the gDNA of the parents. Please see "Identification of the PHACTR2 variation" section in the Results and Supplementary tables 3, 4, and 5 for more details. Furthermore, the possibility of combined variations leading to the disease's phenotype was discussed in the "Discussion" section under the limitation paragraph, lines 385-387.
Reviewer: This does not answer my question. It is highly possible that the proband and the parents have other genetic variations that could also contribute to the phenotype. WES should be done in the proband as well as the parents if the parents’ DNA is available.
Point 2: "No human phenotype was yet associated with any of the PHACTR family members. line 73-73" Is not accurate. Through OMIM, it is found that the phenotype of PHACTR1 is Developmental and epileptic encephalopathy 70".
Response 2: Thank you for this important correction; we changed this sentence in the abstract: "This is the first report of a human variation in this PHACTR family member.". We deleted the sentence in lines 76-77 "No human phenotype was yet associated with any of the PHACTR family members." and replaced it with a corrected one, “PHACTR1 is the only PHACTR family member associated with a human phenotype. Three de novo mutations in the PHACTR1 gene were found to associate with developmental and epileptic encephalopathy (OMIM *608723). “.
Reviewer: This change is acceptable.
Point 3: figure1 D, mouse PHACTR1protein 3D structure was used to predict the impact of the variations of PHACTR2 in human. However, the identity of human and mouse on PHACTR1 is as low as 30%; In contrast mouse PHACTR2 has higher identity with human PHACTR2 (60%). Therefore, it would be better to use mouse PHACTR2.
Response 3: We agree with the reviewer's point that it would have been better to verify the effect of the variation on the PHACTR2 structure and look for its structure in the protein data bank. Unfortunately, there is no experimental structure for PHACTR2 but only for PHACTR1. The confidence of the Alpha fold and Robetta predicted structures of PHACTR2 is low, and they are less reliable than the experimental structure of PHACTR1. Since the variant's location is conserved between PHACTR1 and PHACTR2 around the site, we feel confident about using the structure of mouse PHACTR1. We added a sentence to the manuscript, lines 179-182. "We verified the effect of the candidate variation on the experimental crystal structure of mouse PHACTR1, which is conserved between PHACTR1 and PHACTR2 at the variant location site since there is no experimental structure of PHACTR2."
Reviewer: This is not ideal situation but if there is no PHACTR2 structure available, the description is acceptable although reluctant.
Point 4: The authors identified three missense heterozygous variations with less than 0.5% allele frequencies in the public databases (gnomAD browser, 1000 Ge-128 nomes, ExAC and EVS) in these genes in patient II-3. Taking 0.5% as the cut-off is too high (the standard is too low.) Usually less than 0.1% is more acceptable standard.
Response 4: Indeed, 0.1% is usually the accepted standard to exclude variants by allele frequency in the public database. We used 0.5% allele frequency as a cut-off, so we do not miss any possible pathological variation for two reasons-
- We worked with a Bedouin kindred, characterized by a high frequency of consanguineous marriages; thus, their allelic frequency may not be concurrent with the public databases (gnomAD browser, 1000 Genomes, ExAC and EVS).
- We looked for the possibility that the patient's phenotype may be caused by a combination of heterozygous variations in genes reported having mutations causing familial and non-familial DCM (Supplementary Table 2) along with the homozygous PHACTR2 variant. Therefore, to avoid missing any variation, we used a higher allelic frequency cut-off for the identification of combined variations that might lead to the disease's phenotype.
Reviewer: Albeit the authors explained why they used 0.5% as cut-off, the loosing standard does not meet the usual criteria of genetic variants for pathological significance.
Author Response
Response to Reviewer 1 Comments
Point 1: WES was done only in the proband. It is better to have WES done in the proband and the parents. The identification of variations in PHACTR2 may be combined with variations in other genes as well. This can be excluded with the WES analysis on the parents gDNA.
Response 1: We hypothesized that pathological variations are causing the patient's disease and, indeed, these variations would be inherited from the parents. Accordingly, and instead of trio WES we verified each of the suspected heterozygous/homozygous variations that met the exclusion criteria identified in the patient, in his parents and the healthy siblings. The segregation was confirmed by PCR amplification of the DNA containing the variation, followed by Sanger sequencing, thus enabling the exclusion of variations based on the gDNA of the parents. Please see "Identification of the PHACTR2 variation" section in the Results and Supplementary tables 3, 4, and 5 for more details. Furthermore, the possibility of combined variations leading to the disease's phenotype was discussed in the "Discussion" section under the limitation paragraph, lines 385-387.
Reviewer: This does not answer my question. It is highly possible that the proband and the parents have other genetic variations that could also contribute to the phenotype. WES should be done in the proband as well as the parents if the parents' DNA is available.
Response (round 2): In response to the reviewer's question, unfortunately, the parents' DNA is presently unavailable. However, in order not to miss any possible variation which could contribute to the phenotype, we now verified in the patient's exome all heterozygous/homozygous variations with a frequency of 0% to 0.1% in the public databases (Gnomad, 1000 geomes, ESP 6500, and ExAc). Since the dominant segregation was negated based on the pedigree. We added under the heading "Identification of the PHACTR2 variation", the 2nd sentence “A heterozygous variation could cause DCM in the case of compound heterozygous, a de-novo, or a low-penetrance variation.” (Line 128-129). To identify all heterozygous variations, we proceeded as follows and detailed in the text now: “Second, we proceeded to verify all other possible causative heterozygous variations. We found 8,624 heterozygous variations in the patient exome. Only 135 had an allele frequency of less than 0.1% in the public databases mentioned above and less than 0.5% in our internal Bedouin laboratory Exome database. None of these variations were identified in the same gene, excluding the possibility of compound heterozygous as causative. Assuming a de-novo variation, we negated all variations reported in the public databases- gnomAD and GeniePool, which mostly represent the Bedouins population, remaining with 37 variations. We continued the negation process based on the benign aggregated prediction score by Franklin browser, variations in genes associated with different clinical presentations, and finally, excluded three remaining variations in genes with low expression in the cardiac tissue (Supplementary Table 4). Thus, probably none of these variations represent a de-novo, possibly causative variation.” (Lines 139-150).
As already presented in the Results, all the bi-allelic variations, except in PHACTR2, were negated based on segregation.
We thank the reviewer for this criticism and hope we have succeeded in addressing it properly now.
Point 2: "No human phenotype was yet associated with any of the PHACTR family members. line 73-73" Is not accurate. Through OMIM, it is found that the phenotype of PHACTR1 is Developmental and epileptic encephalopathy 70".
Response 2: Thank you for this important correction; we changed this sentence in the abstract: "This is the first report of a human variation in this PHACTR family member.". We deleted the sentence in lines 76-77 "No human phenotype was yet associated with any of the PHACTR family members." and replaced it with a corrected one, “PHACTR1 is the only PHACTR family member associated with a human phenotype. Three de novo mutations in the PHACTR1 gene were found to associate with developmental and epileptic encephalopathy (OMIM *608723). “.
Reviewer: This change is acceptable.
Response (round 2): We are pleased that the reviewer accepted our change.
Point 3: figure1 D, mouse PHACTR1protein 3D structure was used to predict the impact of the variations of PHACTR2 in human. However, the identity of human and mouse on PHACTR1 is as low as 30%; In contrast mouse PHACTR2 has higher identity with human PHACTR2 (60%). Therefore, it would be better to use mouse PHACTR2.
Response 3: We agree with the reviewer's point that it would have been better to verify the effect of the variation on the PHACTR2 structure and look for its structure in the protein data bank. Unfortunately, there is no experimental structure for PHACTR2 but only for PHACTR1. The confidence of the Alpha fold and Robetta predicted structures of PHACTR2 is low, and they are less reliable than the experimental structure of PHACTR1. Since the variant's location is conserved between PHACTR1 and PHACTR2 around the site, we feel confident about using the structure of mouse PHACTR1. We added a sentence to the manuscript, lines 179-182. "We verified the effect of the candidate variation on the experimental crystal structure of mouse PHACTR1, which is conserved between PHACTR1 and PHACTR2 at the variant location site since there is no experimental structure of PHACTR2."
Reviewer: This is not ideal situation but if there is no PHACTR2 structure available, the description is acceptable although reluctant.
Response (round 2): We are pleased that the reviewer accepted our reply.
Point 4: The authors identified three missense heterozygous variations with less than 0.5% allele frequencies in the public databases (gnomAD browser, 1000 Ge-128 nomes, ExAC and EVS) in these genes in patient II-3. Taking 0.5% as the cut-off is too high (the standard is too low.) Usually less than 0.1% is more acceptable standard.
Response 4: Indeed, 0.1% is usually the accepted standard to exclude variants by allele frequency in the public database. We used 0.5% allele frequency as a cut-off, so we do not miss any possible pathological variation for two reasons-
- We worked with a Bedouin kindred, characterized by a high frequency of consanguineous marriages; thus, their allelic frequency may not be concurrent with the public databases (gnomAD browser, 1000 Genomes, ExAC and EVS).
- We looked for the possibility that the patient's phenotype may be caused by a combination of heterozygous variations in genes reported having mutations causing familial and non-familial DCM (Supplementary Table 2) along with the homozygous PHACTR2 variant. Therefore, to avoid missing any variation, we used a higher allelic frequency cut-off for the identification of combined variations that might lead to the disease's phenotype.
Reviewer: Albeit the authors explained why they used 0.5% as cut-off, the loosing standard does not meet the usual criteria of genetic variants for pathological significance.
Response (round 2): In compliance with the reviewers remark we now changed the cut-off to 0.1%. This was done for both heterozygous and homozygous variations. For the heterozygous variations please see our reply to the 1st point. For the homozygous variations, we changed the text: "Next, we identified 81 homozygous variations with less than 0.1% allele frequencies in the public databases (gnomAD browser, 1000 Genomes, ExAC, and EVS). A total of 69 out of the 81 variations were ruled out by our internal laboratory exome database of the Bedouin population.”. (Lines 153-157). The change of the cut off also resulted in changes in the Supplementary Tables 4 and 5.
- The paper was edited by MDPI language editing services.
An English editing certificate is attached.
Round 3
Reviewer 1 Report
The authors addressed my concerns; I have no more questions.